# Probabilistic Program Induction for Intuitive Physics Game Play

## Abstract

Recent findings suggest that humans deploy cognitive mechanism of physics simulation engines to simulate the physics of objects. We propose a framework for bots to deploy similar tools for interacting with intuitive physics environments. The framework employs a physics simulation in a probabilistic way to infer about moves performed by an agent in a setting governed by Newtonian laws of motion. However, methods of probabilistic programs can be slow in such setting due to their need to generate many samples. We complement the model with a model-free approach to aid the sampling procedures in becoming more efficient through learning from experience during game playing. We present an approach where a myriad of model-free approaches (a convolutional neural network in our model) and model-based approaches (probabilistic physics simulation) is able to achieve what neither could alone. This way the model outperforms an all model-free or all model-based approach. We discuss a case study showing empirical results of the performance of the model on the game of Flappy Bird.

## 1 Introduction

The last few years have been marked with exceptional progress in the field of Artificial Intelligence (AI). Much of the progress has come from recent advances in deep learning. Models employing deep neural networks achieved remarkable performance in many areas including speech recognition, object recognition and reinforcement learning (LeCun et al., 2015; Mnih et al., 2016; 2015; Gu et al., 2016). In reinforcement learning, Mnih et al. proposed a deep learning approach to estimate Q-learning function of state and action tuples in Atari games to achieve human-level performance (Mnih et al., 2015; 2013; Guo et al., 2014). Silver et al. applied a similar approach of deep learning to learn policy and value functions of states and action in the complex game of AlphaGo (Silver et al., 2016).

Another front where AI is progressing exceptionally is approaching AI through drawing inspiration from human cognitive processes (Lake et al., 2016; Kim et al., 2017). Lake et al. developed a model for human-level concept learning through probabilistic induction (Lake et al., 2015). The approach deploys methods of probabilistic programming (Ghahramani, 2015; Goodman & Tenenbaum, 2016) to construct computational frameworks that capture human learning abilities in forming concepts. The construction of computational frameworks was used to draw insights on human cognition in other areas, Battaglia et al. proposed a model based on an intuitive physics engine as a cognitive mechanism humans use to make robust inferences in complex natural scenes (Battaglia et al., 2013; 2016).

In human development, infants have primitive concepts of how objects move in their environments, this is observed through their ability to track moving objects around them. It is through these primitive concepts infants grow to learn faster and make more accurate predictions (McCloskey et al., 1983; Lake et al., 2016). Experiments on humans cognitive processes of intuitive physics inference show that as tasks of inference are harder, the response time of humans increases (Hamrick et al., 2015). This is due to a trade off between response time and number of physics simulations performed, and the harder the task gets, the more simulation runs humans seem to perform.

Several attempts proposed models for agents to develop a sense of intuitive physics that humans possess. Agrawal et al. approached the problem by reverse engineering intuitive physics (Agrawal et al., 2016). Using robot arms, the agent performed enormous number of actions (i.e. pokes) on objects placed on a table to understand the process by how objects move. The approach is inspired by how infants develop their physics intuition. The model is then tested by requiring an agent to move objects on a table to match a given final state of object positions on the table. Wu et al. proposed a model capable of predicting the movement of objects placed on an inclined surface given image pixel data (Wu et al., 2015). The model incorporates deep learning methods to learn physical features of objects and a 3D physics simulation to predict their trajectories and where an object will most likely stop, the approach was capable of achieving an accuracy comparable to human subjects.

In this paper, we propose a framework for bots to deploy tools for interacting with the physics of their environments. The bots employ a coupling of a probabilistic program with a physics simulation engine to do inference of moving objects in a setting governed by Newtonian laws of motion. However, methods of probabilistic programs can be slow in such setting due to their need to generate many samples. Hence we complement our approach with a model free component to aid the sampling procedures in becoming more efficient through learning from experience during game playing. We present a case where a myriad of model-free approaches (CNN in our model) and model-based approaches (probabilistic programming and physics simulation) is able to achieve what neither could alone. The performance of the model outperforms an all model-free or model-based approach (Evans & Grefenstette, 2018). It has been evident that such approaches combine the best of both worlds (Battaglia et al., 2018). The case study shows empirical results of the performance of the model on the game of Flappy Bird, a game of a bird in free fall and required to avoid obstacles by jumping through openings. Our model exhibits similar patterns in behavior of humans when it comes to the trade off between sampling time versus accuracy of decisions (Hamrick et al., 2015) and the ability to learn through experience of game play.

In sections 2, 3 and 4 of the paper, we propose the framework and discuss the process of inference and learning of parameters. Sections 5 include empirical results of the model performance while playing Flappy Bird and discussion.

## 2 Methods

Given the state of an agent in an environment governed by Newtonian laws of physics, the goal is to have an agent that predicts the desired behavior given a state. It does that by probabilistically sampling actions most suitable to achieve the desired behavior. To illustrate, given a bot in a state very close to hit the floor, the agent first should be able to infer that it needs to increase altitude and then probabilistically bias the sampling process to actions that are in line with increasing altitude to avoid collision.

The decision making pipeline of the agent includes two main subparts; the first is a convolutional neural network (CNN) and the second is a probabilistic framework for sampling actions in an intuitive physics setting. The general architecture of the pipeline is included in figure 1. The CNN takes pixel data as inputs and produces the parameter $\alpha$ corresponding to prior probabilities for the probabilistic program. The probabilistic program is parametrized by $\alpha$ the prior probabilities for each action, $\theta$ the probabilities of each action, $a$ the sampled action, $\gamma_a$ the velocity of the action and $c_h$ the collision state at time $h$.

### 2.1 Convolutional Neural Network (CNN)

The objective of the CNN is to make the sampling procedure of probabilistic programs more efficient, this is done through estimate parameters of the prior probability distribution of actions to be taken given a state of the agent. This helps the probabilistic model in sampling more effectively through skewing a Dirichlet distribution in a manner where actions sampled are more likely to match a desired behavior. CNN has a similar architecture to that developed by Mnih et al. (Mnih et al., 2015).The CNN takes the last 4 frames as inputs and outputs $\alpha$ parameterizing the prior for the distribution of actions probabilities. The input of

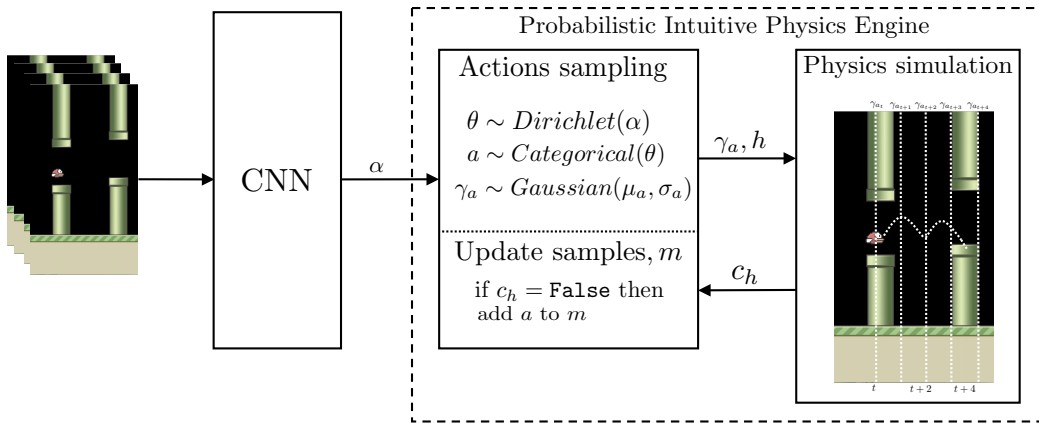

Figure 1: The decision making pipeline, it starts with the CNN having the objective of improving the sampling process through estimating $\alpha$, then a probabilistic intuitive physics engine samples decisions according probabilities estimated by CNN and simulates samples in a physics simulation. The physics simulations returns the state of the agent after a period of time $c_h$ ($c_h =$ False if unwanted collision was not observed). Flappy bird is used as an example here to demonstrate the structure of the model, the results and discussion section include an implementation of the model.

the neural network consists of an $80 \times 80 \times 4$ frames of pixel data for the past 4 time steps after preprocessing. Preprocessing denoted $\phi$ includes transforming the raw pixel data to grayscale then rescaling frame size to a resolution of $80 \times 80$. The first hidden layer convolves 32 filter of size $8 \times 8$ with stride 4 on the input frames and applies rectifier nonlinearity. The second hidden layer convolves 64 filters with sizes of $4 \times 4$ with stride 2 then applies rectifier nonlinearity. The third hidden layer convolves 64 filters with sizes of $2 \times 2$ with stride 1 then applies rectifier nonlinearity. The fourth layer is a fully connected 512 nodes with rectifier nonlinearity. Then the output layer is fully connected and has as many nodes as there are actions in the game. The output of the CNN parametrizes a Dirichlet distribution in the probabilistic framework.

## 3 Learning

The model learns from experience through the data generated while game playing. Sources of the data include positions information of objects in the environment, pixel data of the state, actions taken by the agent and the rewards received at every time step.

The CNN learns from the pixel data of past states and actions; the inputs to the CNN are the pixel data and the outputs are the frequencies of actions in subsequent time steps of a predefined interval. $\alpha_i$ is the frequency of making decision $i$ in the future $\Delta$ time steps after state $s$. The training sample of the model has $(s, \alpha)$ tuples after which the model was negatively rewarded are not included in the training of the CNN since the goal is to learn about values of parameter $\alpha$ resulting in the bot being positively rewarded. The CNN is fit through applying stochastic gradient descent on the following cost function:

$$L(\alpha, \phi, s, \kappa) = \big(\alpha - \mathcal{M}(\phi(s), \kappa)\big)^2$$

Where the parameters for the neural network are denoted by $\kappa$ .

## 4 Inference

We use probabilistic programming to perform the tasks of inference in a fashion similar to the discussion by Gharmani et al. in (Ghahramani, 2015). This section discusses the process

by which the data for the world of the agent and its actions are generated. We employ control flow to sample actions resulting in no collisions.

At every iteration, the agent will use a probabilistic program coupled with the physics engine algorithm illustrated in Algorithm 1 to do inference about the physics of its future. The Algorithm takes as inputs the state of the world defined by the past 4 frames of pixel data and past positions data of objects in the environment. The CNN estimates the direction the agent should be moving towards through estimating Dirichlet parameters $\alpha$. An alternative approach is to is to provide a $\alpha$ for a uniform Dirichlet on probabilities of actions which can be computationally cumbersome under tight time constraints to make a decision.

---

**Algorithm 1** Probabilistic Intuitive Physics Process

---

  **procedure** GETACTION$(s, t, m)$
    Inputs are the state $s$, current time $t$ and previously sampled actions $m$
    Initialize the simulation horizon $h$ to constant $c$
    $\alpha \leftarrow \mathcal{M}(\phi(s))$           ▷ estimate the starting $\alpha$ from the CNN
    **while** elapsed time $< \epsilon$ **do**       ▷ keep sampling for a duration of $\epsilon$ seconds
        $a, c_h \leftarrow$ SAMPLEACTIONS$(s, t, \alpha, h)$
        **if** $c_h$ is False **then**       ▷ observe when unwanted collision didn't occur
            add $a$ to $m$       ▷ store samples resulting in no unwanted collisions

$$\alpha_i \leftarrow \alpha_i + \sum_{j=1}^{h} 1_{a_j = i} \qquad \qquad \triangleright \text{ update } \alpha \text{ parameters}$$

            $h \leftarrow h + \delta$       ▷ expand the horizon of the simulation
        **end if**
    **end while**
    $\tilde{p}(a_t = x | c_h = \text{False}) \leftarrow \sum_{i=1}^{|m|} 1_{m_{i,t} = x}$ ▷ estimating the conditional from samples in $m$
    $\tilde{a}_t \leftarrow \arg\max_{a_t} (\tilde{p}(a_t | c_h = \text{False}))$
    **return** $\tilde{a}_t$
  **end procedure**
  **procedure** SAMPLEACTIONS$(s, t, \alpha, h)$
    $\theta \leftarrow Dirichlet(\alpha)$       ▷ sample $\theta$, the probability of actions
    $a_t, ..., a_{t+(h-1)} \leftarrow Categorical(\theta)$     ▷ sample actions for timesteps $t$ to $t + (h - 1)$
    $\gamma_{a_i} \leftarrow Gaussian(\mu_{a_i}, \sigma_{a_i})$       ▷ sample impact on the velocity
    $c_h \leftarrow$ PHYSICSSIMULATION$(s, [\gamma_{a_t}, ..., \gamma_{a_{t+(h-1)}}], h)$ ▷ simulate sampled plan for next $h$
steps
    **return** $a$, $c_h$
  **end procedure**

---

The agent proceeds with an iterative process between a probabilistic program and a physics engine simulation. The process starts with sampling probabilities for each action denoted $\theta$ from a Dirichlet distribution. The agent samples $h$ actions from a categorical distribution parametrized with $\theta$. To account for the possible stochasticity of actions, a Gaussian distribution is fitted overs the observed change in velocities for every action type, the model learns the distributions through the history of its position data and the actions taken in the past. For every move the agent sampled, $\gamma_{a_i}$ is sampled from a Gaussian corresponding to action type $a_i$.

After an action plan is sampled from the probabilistic program, the simulation engine will simulate the plan. The physics simulation returns the status of the agent after $h$ time steps. The process continues to sample then simulate in an iterative manner for an $\epsilon$ milliseconds. The physics engine estimates physics characteristics of the environment (for example the gravitational acceleration) through the application of Newtonian laws of motion to the historical positions data it observed.

Upon the end of the iterative sampling and simulation process, the bot is to estimate the distribution $p(a_t | c_h = \text{False})$ through the set of simulations of actions called $m$. Given the samples it generated in $m$, the conditional density is given by:

$$p(a_t = x | c_h = \texttt{False}) \propto \tilde{p}(a_t = x | c_h = \texttt{False}) = \sum_{i=1}^{|m|} 1_{m_{i,t}=x}$$

The decision on which action to take at the current time step denoted $\tilde{a}_t$ is then given by:

$$\tilde{a}_t \sim argmax_{a_t} \ \tilde{p}(a_t | c_h = \texttt{False})$$

Before starting a new iteration of sampling new actions then simulating, the parameters $\alpha$ are updated with the actions $a_t, ..., a_{t+(h-1)}$ if a simulation results in no collision (i.e. $c_h = False$). If so, the process increments the simulation horizon $h$ with $\delta$ time steps, the strategy is to continuously expand the simulation horizon as no collisions are observed for the bot to detect potential further away obstacles in its direction. The process of sampling and simulation loops until $\epsilon$ milliseconds pass. The Algorithm returns the action providing the highest probability for the bot to survive unwanted collisions in $h$ time steps.

## 5 EXPERIMENTS AND RESULTS

The approach was tested on the game of Flappy Bird, a game of inference about the physics of the bird and its environment towards avoiding collision with obstacles. A replica implementation of the game is available at on `github` along with an implementation of DQN proposed by Mnih et al. (Mnih et al., 2015; ker, 2016; Chen, 2015). The game is available for play at `http://flappybird.io`. During the game, bird is required to chose from flapping or doing nothing to avoid collisions and pass through the openings.

Figure 2 demonstrates how CNNs aid probabilistic programs in sample more efficiently, the task for the agent is to infer about actions with highest probability in terms of passing through openings facing Flappy Bird. The agent is to sample action $\theta$ from a prior parametrized by $\alpha$, the $\alpha$ is given by the CNN to help reduce the number of samples needed to pass through the opening. Given actions of flap and do nothing, with $\theta^*_{flap} = 0.072$ the agent is more likely to sample flaps in its plan as to maintain its current height moving towards an opening. Figure 2(a) illustrates the frames of pixel data used as inputs to estimate the Dirichlet (i.e. Beta since two actions are offered for bird to pick from), the probability of $\theta > \theta^*_{flap}$ is larger resulting in simulating more samples with ascending altitude. On the contrary, in (b) the sampling distribution is skewed towards values less than $\theta^*$ resulting in the sampling process halting from flapping in most of its samples. Hence, the CNN skews the Dirichlet distribution in a manner where actions sampled match the desired change in altitude.

Figure 3 demonstrates the performance of the model against alternative approaches and state of the art techniques. Average scores are calculated after running each trained model for 10 times and observing the final score. PB-CNN is the proposed methodology and PB-Uniform is an alternative approach where a CNN is not present, instead the Dirichlet distribution is parametrized with all one parameter resulting in a uniform distribution over the parameter $\theta$. DQN is an implementation of Mnih et al. (Mnih et al., 2015) on Flappy Bird. Human data are gathered through players on the web page in (hum, 2017).

Figure 3 (a) shows the average accuracy of PB-CNN and PB-Uniform for different $\epsilon$ milliseconds of time allowed for iterative sampling and simulation process. The advantage CNN brings to the model is significant, this is because CNN narrows down the sampling space significantly allowing the model to explore the conditional distribution of samples given no collisions much faster (i.e. higher frequency of samples resulting in $c_h = False$). In figure 3 (b) we show the average score per training epoch of the DQN approach. In figure 3 (c), the average score of humans play is close to PB-Uniform with $\epsilon = 30$ms and $\epsilon = 40$ms. After training the CNN part of PB-CNN of the model on 10000 frames of game play, the model's performance improves significantly. The performance of PB-CNN with $\epsilon = 1.5$ms is similar to DQN in average score. The average score is calculated for 10 times of game play.

The average score for humans was 11.27 in 47 million games played, 95% of them had a score of 6 points or lower (hum, 2017). One possible reason why humans under-perform could be explained by their significantly large delay in response time compared to methods

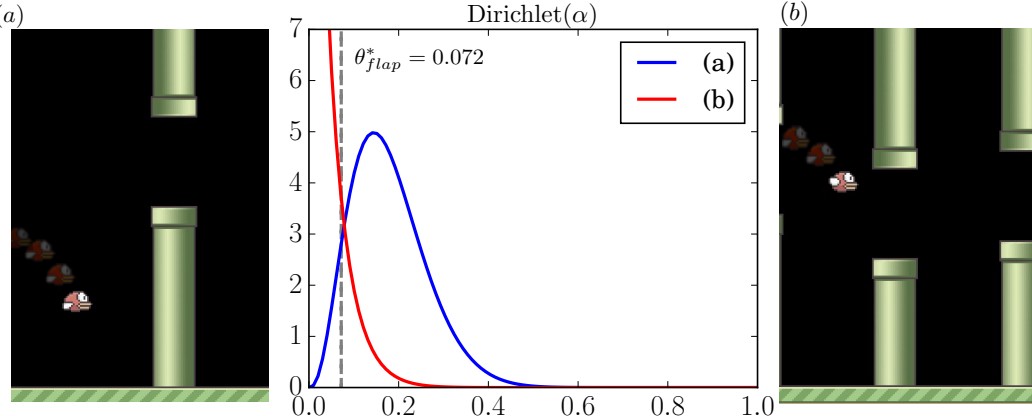

Figure 2: This figure demonstrates how CNN helps making the process of sampling more efficient. Frames from the game are inputs to the CNN which outputs $\alpha$ parameters for the Dirichlet distribution illustrated in the figure for the case of Flappy Bird. The dashed line of $\theta^*_{flap} = 0.072$ corresponds to the probability of flap where bird maintains the same height. In (a) bird is descending and is faced with a gap that is higher than its current altitude, a trained CNN outputs $\alpha = (3.6, 16.8)$ biasing the sampling procedure to generate actions that will more likely result in its ascending towards the opening. In (b) bird is descending where a flap will result in its collision, the CNN outputs $\alpha = (0.7, 19.7)$ corresponding to values of $\theta$ that will more likely result in its descending towards the opening.

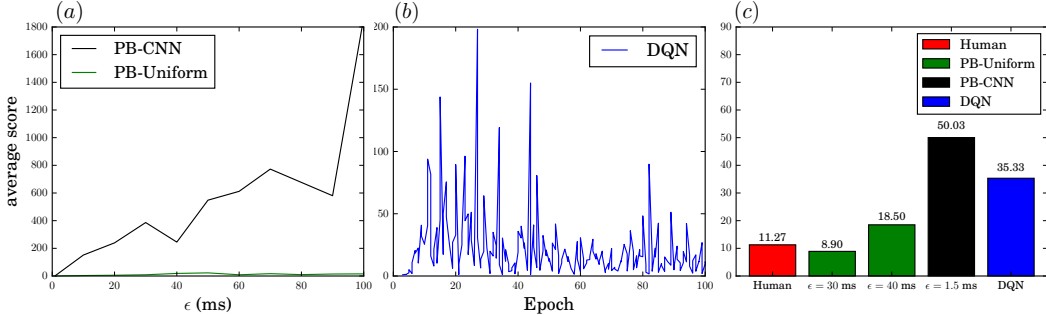

Figure 3: Results of models trained to play Flappy Bird. PB-CNN is the proposed model, PB-Uniform is composed of the model proposed without the CNN where the probabilistic program starts with an all-ones $\alpha$ parameter. DQN is the model proposed by (Mnih et al., 2015), human data for game play are reported by the web page in (hum, 2017)

discussed in the paper here. Under a task of inference about the physics of the world, research suggested that humans response rate was in between 500 and 2000 milliseconds depending on the hardness of the inference task at hand (Hamrick et al., 2015). The urgency of making a decision leaves no ample time for sampling and simulating the physics of the game that could be enough for humans to perform as well as bots.

## 6    CONCLUSION AND FUTURE WORK

We propose a framework for bots to maneuver in environments of intuitive physics inspired by cognitive processes of humans. The approach draws inspiration from recent approaches of modeling concept learning where the framework includes a coupling of a probabilistic program with a physics simulation engine. The model was tested on the game of Flappy

Bird and compared to state of the art techniques of model-free approaches. Advantages of our model over model-free approaches namely DQN is the ability to learn from very few examples relative to the number of examples DQN requires. Advantages of our model over model-based approaches is its ability to learn from experience.

Potential future work include investigating approaches to learn about rewards structures in games of physics intuition. This will enable bots to perform more complex moves beyond simpler tasks such as the ones in the illustrated game of Flappy Bird where the objective is to avoid unwanted collision. Other games such as Space Invaders involve learning strategies of shooting and hiding that are beyond the capabilities of the model in its existing state. Another potential future direction is to deploy neural network to detect objects in the frames of the game rather than explicitly having access to position data of the objects in the game. This would potentially help the framework better generalizes over other games.

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
