# OpenReview forum: "Probabilistic Program Induction for Intuitive Physics Game Play"
_ICLR.cc/2019/Conference_

### Official Review · AnonReviewer3 · 2018-10-18
**Writing and results should be improved**

**Rating:** 2
**Confidence:** 4

**Review:**

This paper solves Flappy bird by combining DQN and probabilistic programming. I think this is in general a good avenue to explore.

However I found the paper to be poorly written. For example, notation is not properly introduced, there are many mathematical mistakes and typos in the written text and citations. This makes it very hard to understand what is actually going on.

It is also not clear what is the probabilistic program and what are we conditioning on? What is the inference algorithm? Maybe it's useful to expand more on how this ties to the "RL as inference" framework (see e.g. Levine, 2018). It seems like we are doing rejection sampling where the condition is "no collision". As a result, I'm not sure whether sampling from prior is a competitive baseline.

For the DQN experiment, the learning curve seems very noisy in a way that it's unclear whether a fair conclusion can be drawn only from one run (as it appears to be done).

The experiments also feel a bit contrived to make a strong case for probabilistic programming + DQN.

---

### Official Review · AnonReviewer1 · 2018-11-02

**Rating:** 4
**Confidence:** 2

**Review:**

This paper combines a model-free approach and a model-based approach for the game of Flappy Bird. The model-free approach is a CNN on the screen snapshots, in the same fashion as DQN was used for Atari games. The model-based approach is a probabilistic model of Newtonian laws of motion.
The combination of model-free and model-based approaches is definitely one very relevant issue in machine learning, especially in interactive situations, such as reinforcement learning and robotics. The ideas that this paper combines are state-of-the-art and hence is it informative to see how these two particular techniques from each paradigm work together. The most interesting part is that the probabilistic model restricts the CNN restricts the sample action for the model.
I’m not sure how novel this particular combination is, but other systems have also used a model (or a solver) to restrict the possibilities of the exploration. For instance, AlphaGo combines the model-free approach with the rules of the game (at least in the first versions of AlphaGo), and others use Montecarlo Tree Search in a similar vain.
The paper is generally well-written, with some typos occasionally. However, I think that some parts of the process are no well explained, or explained in the wrong order.
For instance, the title and the abstract misled me for quite a while. The title says program induction, and this is then said to be in a fashion similar to Ghahramani 2015, but no further details are given. Is this using Julia? In any case, where is the induction? Later on, it is said that “the model learns the distributions”. but the exact way is completely missing. In other words, the model-based part is not described and encapsulated in a cryptic PHYSICSSIMULATION. In any case, if the model just learn the parameter, I wouldn’t call this “program induction”, at least in the same way as Lake and others use it, or in the way it is used in the area of “inductive programming”.
The game should be describe at the start as “unwanted collisions” are meaningless for a reader who doesn’t know the goal of the game (which is explained in the last paragraph of the paper).
The main problem is that the physics simulation is not learnt and hence rewritten for other games with other physics. This should be solved, in order to see significant progress for benchmarks such as ALE (and properly compare with DQN and many other variants). Perhaps robotics is a better application area, as the physics are always the same (true physics).
Regarding the experiments, they are not very conclusive, especially because the difference is not that large and it is only one single game. The application to other games would be needed, especially if the physics is different. Also, the parameters are different (the ms) and I didn’t understand if these are the best choices, or the possible choices given the computational limitations. In other words, I don’t know if all techniques are compared in the same compute and data conditions. In Figure 3, the yaxis should be explained.
(hum 2017) I imagine that this refers to the authors, otherwise it is a typo.
I didn’t understand the future work. I didn’t parse the bit: “learn about rewards structures in games of physics intuition”.  Actually, the final paragraph gives hints about how much assistance and specialisation the approach is given and hence the limit of generalisations to other problems. These limitations should be stated from the beginning.
Pros:
-	Important integration of model-free and model-based approaches.
-	State-of-the-art techniques

Cons:
-	I don’t see program induction, despite being in the title.
-	Experiments are limited to one game
-	The physics engine is specialised for this game, and hence the approach is difficult to generalise (automatically) for a range of problems where the models should be different.
Typos:
Gharmani -> Ghahramani
overs -> over
bird is required to chose -> the bird is required to choose

---

### Official Review · AnonReviewer2 · 2018-11-02
**Unconvincing Results**

**Rating:** 3
**Confidence:** 4

**Review:**

The authors present an algorithm that incorporates deep learning and physics simulation, and apply this algorithm to the game Flappy Bird.  The algorithm uses a convolutional network trained on agent play to predict the agent’s own actions given a sequence of frames.  Using this action estimator output as a prior over an action distribution (parameterized by a Dirichlet process), the algorithm iteratively updates the action by rolling out a ground-truth physics simulator of the environment, observing whether this ground-truth simulation yields negative reward, and updating the action accordingly.

While I find the authors' introductory philosophy largely compelling (it draws inspiration from developmental psychology, learning to model the physical world, and the synthesis of model-based and model-free learning), I have concerns with most other aspects of the paper.  Specifically, here are a few points:

1)  The authors only apply their algorithm to a single game (Flappy Bird), a game that has no previously established benchmarks.  In fact, while there is no prior work in the literature on this game (perhaps because it is considered very easy), some unofficial results suggest that it is solvable by a straightforward application of existing methods (see this report:  http://cs229.stanford.edu/proj2015/362_report.pdf).  The authors do apply one baseline (out-of-the-box DQN) to this game, but the reported scores are suspiciously low, particularly in light of the report linked above.  No training curves or additional baselines are shown, and no prior work on this game in the literature exists to compare against.

2)  The authors’ algorithm uses privileged information which eliminates the possibility for a fair comparison to baselines.  Specifically, their algorithm uses ground-truth state (not just image input), and a ground-truth physics simulator (which should be an enormous advantage).  Their one baseline (DQN) does not have either of these sources of privileged information, hence cannot be a fair comparison.

3)  The authors’ algorithm is not general-purpose.  Because the algorithm itself uses a ground-truth environment-specific state, a ground-truth environment-specific simulator, and relies on a “crash boolean” (whether the bird hit a tree) specific to this game, it cannot be applied out-of-the-box on a different environment.

4)  The authors make some claims that are too strong in light of the reported results.  For example, they claim that “the performance of the model outperforms all model-free and model-based approaches” (section 1), while they do not even compare against any model-based baselines (and only a single model-free baseline, DQN, which is not state-of-the-art anymore).

Overall, I would recommend the authors choose a game or set of games that has/have established baselines in the literature, come up with a general-purpose algorithm which doesn’t rely on a ground-truth physics simulator, and more rigorously compare to existing methods.

---

### Meta-Review · Area_Chair1 · 2018-12-15
**Not enough supporting evidence**

**Confidence:** 4
**Recommendation:** Reject

**Metareview:**

The paper presents the combination of a model-based (probabilistic program representing the physics) and model-free (CNN trained with DQN) to play Flappy Bird.

The approach is interesting, but the paper is hard to follow at times, and the solution seems too specific to the Flappy Bird game. This feels more like a tech report on what was done to get this score on Flappy Bird, than a scientific paper with good comparisons on this environment (in terms of models, algorithms, approaches), and/or other environments to evaluate the method. We encourage the authors to do this additional work.